# Spider Mites Detection in Wheat Field Based on an Improved RetinaNet

**Denghao Pang** [1,2], **Hong Wang** [1], **Peng Chen** [1,2,*] **and Dong Liang** [1,2]

1　School of Internet, Anhui University, Hefei 230601, China
2　National Engineering Research Center for Agro-Ecological Big Data Analysis & Application, Anhui University, Hefei 230601, China
*　Correspondence: pchen@ahu.edu.cn

**Abstract:** As a daily staple food of more than one third of the world's population, wheat is one of the main food crops in the world. The increase in wheat production will help meet the current global food security needs. In the process of wheat growth, diseases and insect pests have great influence on the yield, which leads to a significant decline. Wheat spider mites are the most harmful to wheat because they are too small to be found. Therefore, how to use deep learning to identify small pests is a hot spot in modern intelligent agriculture research. In this paper, we propose an improved RetinaNet model and train it on our own dataset of wheat spider mites. Firstly, the wheat spider mites dataset is expanded from 1959 to 9215 by using two different angles and image segmentation methods. Secondly, the wheat spider mite feature detection head is added to improve the identification of small targets. Thirdly, the feature pyramid in FPN is further optimized, and the high-resolution feature maps are fully utilized to fuse the regression information of shallow feature maps and the semantic information of deep feature maps. Finally, the anchor generation strategy is optimized according to the amount of mites. Experimental results on the newly established wheat mite dataset validated our proposed model, yielding 81.7% mAP, which is superior to other advanced object detection methods in detecting wheat spider mites.

**Keywords:** wheat spider mites; improved RetinaNet; object detection; image processing

## 1. Introduction

Wheat is one of the major food crops in China, and the guarantee of wheat yield and harvest is crucial to food security and food consumption. During the growth and development of wheat, the frequent occurrence of diseases and insect pests will seriously affect the production of wheat [1]. For example, aphids, suckers, mole crickets, larvae and needle worms attack different parts of wheat at different times, causing serious damage [2,3]. Among them, wheat spider mites have the most serious effect on wheat yield. In addition, wheat spider mites are harmful to crops such as barley, peas and rape, as well as wheat. According to the survey, in 2020, the occurrence area of wheat spider mites in China reached 20.15 million acres, and it is increasing year by year. Although the average insect population ranges from 3 to 25 insects per foot of market length in low-growing areas, if left unchecked, insect populations can proliferate and cause irreparable damage to wheat [4]. Therefore, timely and accurate identification is very important for the control of wheat spider mites.

On the one hand, however, mite outbreaks tend to be severe, sporadic and often blindsided, making scientific application and precise control very difficult. Wheat spider mites, on the other hand, tend to be very small and difficult to see with the naked eye, making accurate detection of very small objects like wheat spider mites a challenging problem. At present, manual identification is the most commonly used method to detect wheat spider mites infestation [5]. However, due to the small size and density of wheat

spider mites, manual identification efficiency is low. Therefore, timely, scientific, accurate and effective detection and control measures are of great significance to reduce wheat production loss and increase farmers' income. With the extensive research of a large number of scholars on computer vision and image processing technology, image based pest recognition has become a feasible solution. Although the technique saves a great deal of time and effort compared to manual work, accurate identification remains challenging. First, it is very difficult to distinguish wheat spider mites from the background because the camera takes pictures with complex background. In addition, the camera usually takes pictures with a high resolution, which is a difficult problem for quick detection. Moreover, the Angle and illumination of the shot can also affect the quality of the image, increasing the difficulty of recognition. Therefore, developing an effective and accurate method to identify wheat mites is difficult and challenging.

Recently, with the research and development of deep learning in the field of object detection, many agricultural scholars also try to extend the method of object detection to crop disease recognition [6–8], weed detection [9–11], fruit detection and counting [12–14] and other practical applications. Certainly, traditional machine learning has also done excellent work on crop disease recognition. Nazari et al. proposed ANFIS classifier to identify Alternaria disease and leafminer based on color and texture features. Finally, the segmentation and classification accuracy of PlantVillage dataset are 90% and 98%, respectively [15]. However, compared with the pretreatment and other operations of machine learning, convolutional neural network, as a representative of deep learning, can complete end-to-end learning and achieve more excellent results.

Based on Faster-RCNN, He et al. [16] designed a two-layer detection network algorithm to detect brown rice planthoppers(BRPH). The first layer is responsible for detecting the original image and saving the image of the detection target area. The second layer performs object detection on the saved image. The accuracy and recall rate of the algorithm are 94.5% and 88.0% on more than 200 test images, respectively. For large-scale multi-species pest data, Wang et al. [17] proposed a two-stage cascading pest detection method (STOMACH) based on mobile vision. Firstly, the multi-scale context information of images is extracted and a context-aware attention network is established to perform initial classification of crop categories. Then, a multi-projection pest detection model (MDM) was proposed, and crop related pest images were used for training. Finally, attention mechanism and data enhancement techniques were used to improve the effectiveness of pest detection in the field. Tassis et al. [18] first performed instance segmentation with Mask-RCNN, then did semantic segmentation using UNet and PSPNet, and finally did classification and severity estimation using Resnet, which achieved high accuracy in the dataset of coffee trees. Although the above methods have improved the ability of pest identification to a certain extent, they all divide the detection process into multiple stages and do not directly improve the model itself, so there are some limitations [19,20].

For small agricultural pests, Dong et al. [21] proposed an end-to-end model CRA-Net, which contains a channel recalibration feature pyramid network (CRFPN) to improve channel-level feature fusion and an adaptive anchor (AA) module to predict arbitrary shaped anchors. Outperforming other state-of-the-art models, the novel CRA-Net achieved a average precision (AP) of 67.9%. In [22], Lin et al. introduced an adaptive feature fusion into the feature pyramid network to extract more features of the pest. Then, they developed an adaptive enhancement module to reduce the information loss of the feature maps. Finally, they constructed a two-stage region-based convolutional neural network and achieved 77.0% accuracy on the AgriPest21 dataset. In order to improve the accuracy of small object detection, Lim et al. [23] proposed a object detection method based on context. The approach used additional features from different layers as contexts by connecting multi-scale features. And they also proposed an object detection with attentional mechanism, which can focus on the target in the image and can contain contextual information from the target layer. For the $300 \times 300$ input, the mean average precision (mAP) of 78.1% was achieved on the PASCAL VOC2007. Zou et al. [24] integrated a channel attention network

to selectively boost the useful features and restrain the useless ones, proposed a dense feature fusion network to improve the sensitivity for small objects, and designed a rotation anchor strategy for reducing the redundant detection regions. Liu et al. [25] proposed a high-resolution detection network (HRDNet), a multi-depth image pyramid network (MD-IPN) and a multi-scale feature pyramid network (MS-FPN) to maintain multiple position information and reduce the information imbalance. Through the optimization of the model, the above methods have achieved certain results and have high reference value.

In addition, as shown in Figure 1, in the actual detection process, there may be many factors affecting the detection results, such as complex or simple background, large or small pest objects, dense or sparse pest distribution, front or side shooting angle, etc. Different environments often lead to wide variations in test results. Mohanty et al. [26] used the same model to conduct experiments on datasets with simple background and complex background respectively. The former has a high accuracy, while the latter has a relatively low accuracy. It is proved that the detection accuracy is greatly limited by illumination condition, background condition and shooting effect. Therefore, real pest dataset are better than laboratory ones for simulating the growing environment of crops and verifying the robustness and identification of models [27–29].

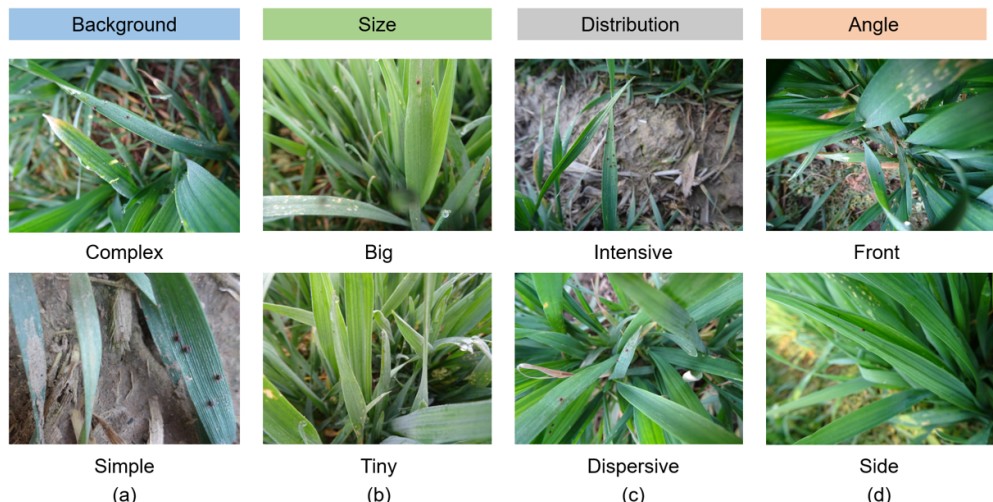

**Figure 1.** Images of wheat spider mites taken under different shooting conditions: (**a**) different backgrounds; (**b**) different sizes; (**c**) different distributions; (**d**) different angles.

In this paper, the RetinaNet model is improved to detect and locate wheat spider mites accurately. Firstly, the feature maps with large resolution can be utilized more fully based on retaining the existing the feature information. Then, the high resolution and low resolution feature maps are fused to extract multi-scale information. Finally, the optimization improves the anchor generation strategy to better match the tag information. Experimental results show that our method performs more effective and outperforms other state-of-the-art methods. The contributions of our method can be summarized as follows:

- A dataset of nearly 2000 wheat spider mite images is constructed through field photography and labeling, and the dataset is extended to 9215 images through data enhancement and image segmentation.
- For the detection of wheat spider mites, we add a detection head specifically for small object in FPN and improve the pyramid structure to obtain more information.
- The anchor generation strategy is optimized and enhanced to improve the detection effect of tiny wheat spider mites.
- Extensive experiments have verified the effectiveness of the improved model and image split, and the mAP has been improved from 63.6% to 81.7%.

## 2. Materials and Methods

### 2.1. Image Acquisition

In order to obtain the real scenes of in-field wheat spider mites, we took nearly 2000 high-resolution images from wheat fields in northern China using cameras or mobile phones. Most image sizes are (5184 × 3888). In addition, to make the dataset more heterogeneous, we took pictures under different conditions. As shown in Figure 1a, the backgrounds of the wheat spider pictures are different, some are more complicated, while some are more simple. Also, the size of wheat spider mites is greatly different. Specifically, larger wheat spider mites have around 100 × 100 pixels and smaller ones have 20 × 20 pixels or less, as can be seen in Figure 1b. As shown in Figure 1c, the distribution of wheat spider mites varies greatly, with some dense distribution and some sparse distribution. To avoid homogeneity, we take photos wheat spider mites from different angles, as can be seen in Figure 1d.

In addition, due to the complex environmental background of wheat spider mites in the field, the image of leaves blocking the object appears in the image collection. However, we did not delete the occluded images, because these occlusions would increase the difficulty of detection, but also train the robustness of the model. Finally, to verify the effectiveness of the proposed method, 70% of the samples are randomly selected as the training set, 20% as the validation set, and 10% as the test set. The number of partitioned data sets can be shown in the first row of Table 1.

**Table 1.** The specific numbers of the original dataset and enhancement dataset.

| Type | Train Set | Val Set | Test Set | Total Numbers |
|------|-----------|---------|----------|---------------|
| Initial | 1371 | 391 | 196 | 1959 |
| Aug | 2675 | 391 | 196 | 3262 |
| Aug + split | 6533 | 1843 | 839 | 9215 |

### 2.2. Dataset Labeling and Enhance

For the expanded dataset, we used Labellimg (https://github.com/tzutalin/labelImg, accessed on 1 November 2022) to mark the images in PASCALVOC format. In addition, sufficient results show that using data enhancement to extend the training set can improve the robustness of the model and prevent overfitting [30,31]. As shown in Figure 2, image enhancement is performed using two different angle schemes. One is to use general operations such as cropping, rotation, color transformation, etc., and the other is to randomly copy the wheat spider mites in the image to other locations in the image to increase the number of objects.

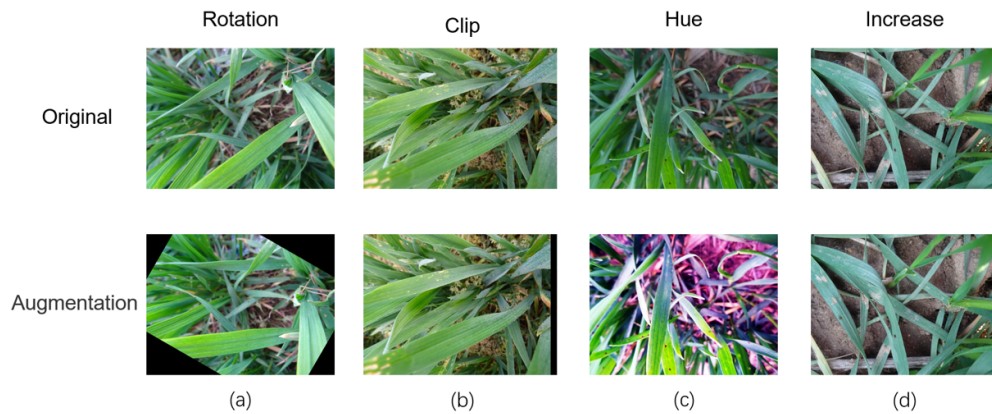

**Figure 2.** The examples of different data enhancement methods: (**a**) rotation; (**b**) crop; (**c**) color; (**d**) copying wheat spider mites.

In Figure 2a–c are the effects of rotation, crop, and hue reversal operations, respectively, while Figure 2d is the effects of wheat spider mites copying operations [32]. The second row of Table 1 shows the number of dataset with both enhancement strategies applied. However, due to the large pixel size of the image and the small pixels of the wheat spider mites, the proportion of the wheat spider mites in the image as a whole is very small. In addition, it is also necessary to take into account the information loss caused by the further compression of image size during the network training. This makes the pixel size of the wheat spider mites so small that it is unrecognizable to the naked eye, which is a major difficulty in the detection of wheat spider mites. Therefore, as shown in Figure 3, we segment each large image (5184 × 3888) into six small images (1999 × 1999) with overlapping edges [33].

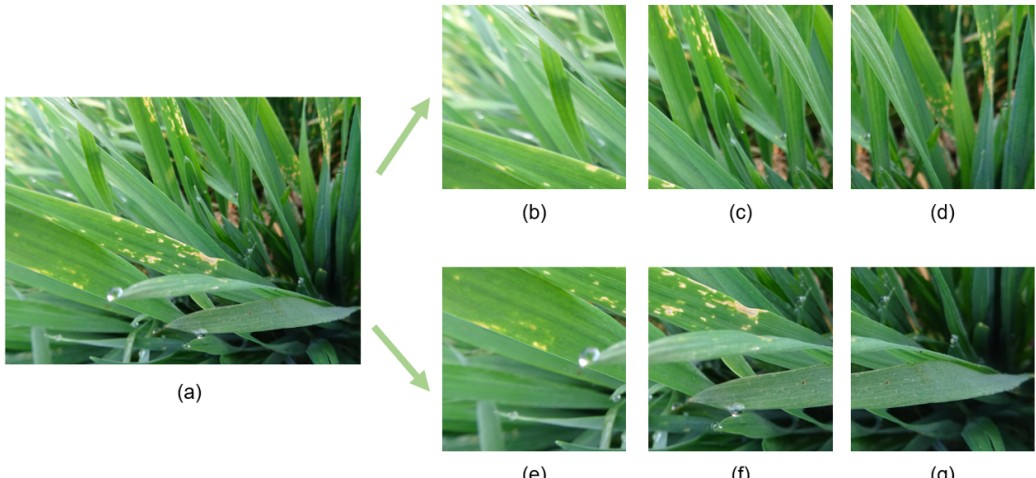

**Figure 3.** The example of image segmentation: (**a**) original image; (**b**–**g**) segmented ones.

On the one hand, the method can alleviate the severe information loss during the image compression to a large extent. On the other hand, the proportion of wheat spider mites in the whole image can be increased, which is conducive to the detection and training of the network. It's worth noting that not every small image contains a wheat mite, so we eliminate these images without objects. However, the problem is that the dataset isn't as big as it was when it first started. Therefore, we need to rebalance the dataset. Specifically, we re-segmented the dataset on the premise that both the original and enhanced images were in the training set. The number of final dataset is shown in the third row of Table 1.

## 3. Network Model

### 3.1. Overview of RetinaNet

Our model is modified based on RetinaNet [34]. RetinaNet is a simple and practical one-stage detection model that can reach or exceed the accuracy of two-stage detector. The structure of it is shown in Figure 4a, which is mainly composed of Bcakbone, FPN and Detector Head.

Backbone: RetinaNet uses ResNet, the most popular classification model, as the backbone network [35]. ResNet consists of four stages, each consisting of a number of cascading residual structures. The residual module consists of three convolutions, a $1 \times 1$ down-sampling convolution for dimensional compression, a $3 \times 3$ spatial convolution for feature extraction, a $1 \times 1$ up-projection convolution for dimensional recovery, and an additional jump connection between inputs and outputs. In addition, the feature map resolution of the four stages is from large to small, and the channel dimension is correspondingly from small to large. Thus, ResNet can generate four feature maps in the form of pyramids.

FPN: The FPN module receives the last three feature maps from the backbone and outputs the last five feature maps by up-sampling and lateral concatenation. Specifically,

the classification information is added by sampling the last three layers and adding them to the previous layer. The last feature image is then continuously compressed and convolved to obtain two feature images with a smaller resolution.

Detector Head: The head module receives five feature maps from the FPN. For each feature map, the head module performs two tasks: category identification and position detection, but the parameters are not shared between the two branches of the head module. Each pixel of the texture map generates a category prediction and four anchor regression results. While the weights of classification and regression branches are not shared, the weights of the head module are shared across the five output feature maps.

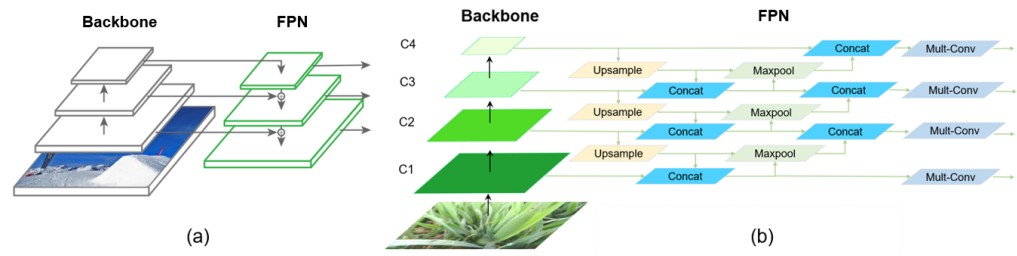

**Figure 4.** Original model and improved model structure: (**a**) FPN structure of RetinaNet; (**b**) our improved RetinaNet.

### 3.2. Small Object Head

Although we have segmented the image to some extent to alleviate the problem that the wheat spider mites in the image is too small, there are still many wheat spider mites of small size, which account for only a small percentage in the image and have very few pixels. Since the detection head of RetinaNet works well for large objects, the detection of small objects such as wheat spider mites is often ignored. Therefore, we have made some improvements to the original detection head. Specifically, we add a tiny wheat spider mite detection predictive head to the original five wheat spider mite detection heads to detect wheat spider mites with a width and height of less than 30 pixels. The new detector utilizes the first-layer feature map of the backbone, which has clearer regression information. This means it is more sensitive to tiny objects, which can help detect polar wheat spider mites. Combined with the other five detection heads, six detection heads are obtained at the FPN stage. Despite the increased compute volume and memory consumption, the increase in the detection heads results in higher detection performance.

### 3.3. Context Fusion

Generally speaking, the shallow feature maps are used to detect small objects because the shallower feature maps have a greater resolution but a smaller field of perception; Conversely, the deeper feature maps have lower resolution but a higher field of perception, so the deeper feature maps are used to detect large objects. This means that shallow feature maps have weak semantic information, but contain stronger feature information, such as shape, texture, edge, etc., while deep feature maps contain strong semantic information. The original FPN structure adds semantic information to the upper feature map by up-interpolation sampling, facilitating category recognition. Shallow feature mapping feeds information back to deep feature mapping by down-sampling to compensate for positional information, which is conducive to the localization of the objects. In addition, shallow and deep feature maps are fused using multi-scale convolutional modules. Specifically, in the shallow layer of the feature map, the convolutional kernels of $3 \times 3$, $5 \times 5$, and $7 \times 7$ are applied to extract the features of different scales, and then stitch the features of different scales together, effectively increasing the perceptual field of the model. More intuitively, as shown in Figure 4b, the original FPN structure is on the left and our improved one on the right.

### 3.4. Improve Anchor Scales

For the original detection head, the detection range is in the range of [8 × 8–128 × 128] relative to the pixel scale of the input image, which is friendly for detecting larger objects, but still large for wheat spiders. To get more accurate information of the wheat mite pixels, we calculated the Ground Truth (GT) size of all images. As shown in Figure 5, most of the width and height are within 50, of which 30–40 is the most, and the proportion within 30 is also very large.

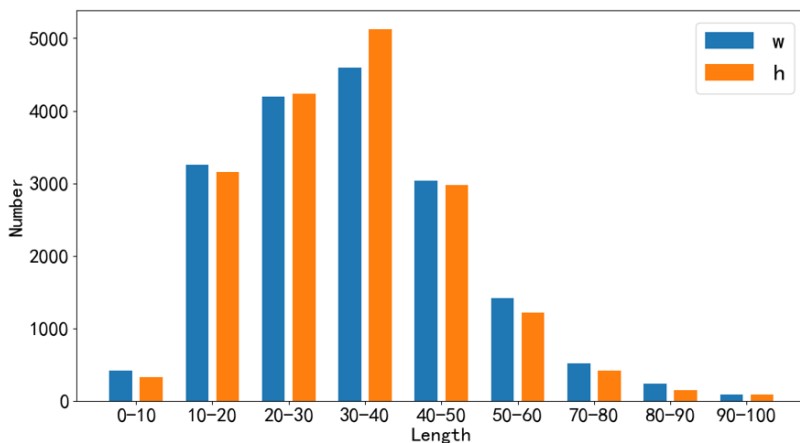

**Figure 5.** Size distribution map of wheat spider wites in our dataset.

This means that existing strategies for generating anchors do not apply to them. In addition, the resulting visualization of the anchors is on the far left of Figure 6.

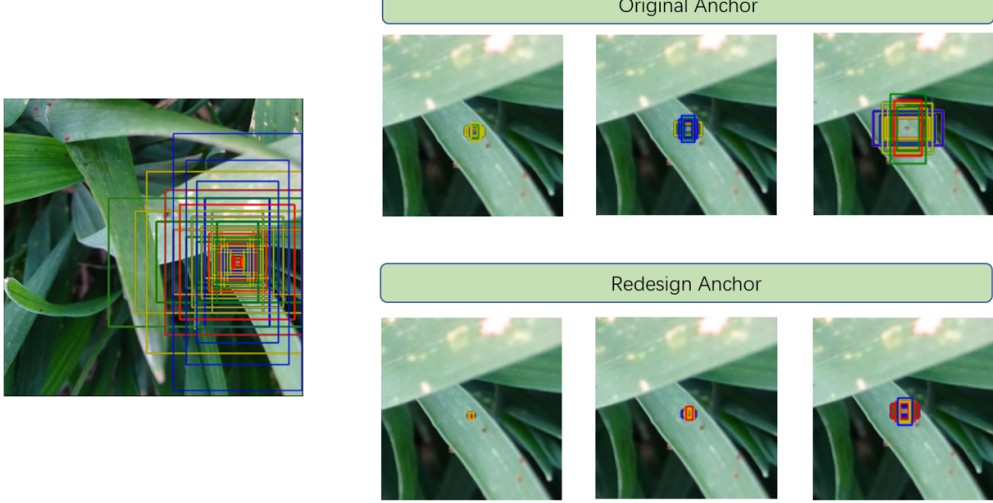

**Figure 6.** Original Anchor vs. optimally designed Anchor: the above are the original three Anchor modules with the smallest scale; the below are the three Anchor modules with the smallest scale after improved.

The lines in the image above represent the separation of the three smallest scales of anchors. As we can see, even the smallest anchor is too large for the smaller mites to successfully match to the GT, so we improve the generation strategy of anchors. Anchor generation can be expressed by the following formula:

$$S_i = \alpha * B * R * S * [2^{\frac{1}{i}}, 2^{\frac{2}{i}} \dots 2^{\frac{i}{i}}], \tag{1}$$

where $\alpha$ is a hyperparameter, $B$ represents the compression scale of the feature maps, $R$ is the aspect ratio generated by the anchor, $S$ represents the cardinal multiplier of the anchor, and $i$ is the scale number. After modifying the scales, the three smallest scale anchors generated are shown in Figure 6. It is not difficult to see that the wheat spider mites that were not matched before can now be matched.

## 4. Experiment

### 4.1. Experiment Setting

In our experiments, we use PyTorch framework and MM-Detection framework to train the proposed model. The experimental environment is Ununtu 18.04 operating system running on a server with an Intel(R) Core i7-12700K CPU 3.60 GHz and two NVIDIA RTX3080Ti (12G) GPUs. In addition, the software we used is JetBrains PyCharm Community Edition 2019.2.6 (64). Our experimental parameters are set according to the convention. Specifically, on Epoch 8 and 11, using pre-trained ResNet as the backbone and SGD as the optimizer, the momentum is set to 0.9, and the learning rate is reduced to one tenth of the original. In particular, our 12G GPU memory makes the Batchsize set to 4 at most, so the initial learning rate is set to 0.0025 accordingly.

### 4.2. Model Evaluation Metrics

To evaluate the performance of the model, we use the inference time, parameters, $R$ (recall rate) and mAP (mean Average Precision) as evaluation metrics. The inference time and number of parameters can measure efficiency and computation. Recall and mapping formulas are the following Equations (2) and (4):

$$\text{Recall} = \frac{TP}{TP + FN}, \tag{2}$$

$$\text{AP} = \int_0^1 P(r)dr, \tag{3}$$

$$\text{mAP} = \frac{\frac{1}{n}\sum_{i=1}^{n} AP_i}{n}, \tag{4}$$

where $TP$ represents the number of positive cases correctly classified as positive cases, $FN$ represents the number of positive cases incorrectly classified as negative cases, $n$ is the number of classes, and $n = 1$ in this case. In the field, it is best to miss the correct targets as little as possible. Therefore, according to the formula, a higher recall rate means that we miss fewer targets. mAP is the most commonly used index in target detection and evaluation, which can measure the effect of positioning and classification at the same time.

### 4.3. Comparison with Other Models

On the wheat mite dataset, the improved RetinaNet was compared with other classical models, and the experimental results are shown in Table 2.

**Table 2.** The results of different models on our dataset.

| Model | Backbone | Inference Time (s/iter) | Params (M) | Recall (%) | mAP (%) |
|---|---|---|---|---|---|
| SSD-300 | VGG16 | 0.098 | 23.75 | 87.0 | 62.1 |
| Yolo-v3 | DarkNet53 | 0.192 | 61.52 | 80.9 | 75.9 |
| Faster-RCNN | ResNet50 | 0.183 | 66.67 | 88.3 | 77.3 |
| RetinaNet | ResNet50 | 0.168 | 41.02 | 88.9 | 77.4 |
| Cascade-Rcnn | ResNet50 | 0.241 | 75.48 | 83.9 | 78.4 |
| RetinaNet-improved (ours) | ResNet50 | 0.269 | 63.31 | 90.2 | 81.7 |

The values in bold indicate the best results. The improved model has the highest recall rate and mAP compared to other models, while also comparing their inference times and

number of parameters. For example, our model increased the recall rate by 1.9% and mAP by 4.4% with fewer parameters compared to Faster-RCNN [36]. Of course our Inference time is a bit long due to the adding of high-resolution detection feature maps for small targets in our model. Obviously, higher resolution means more parameters. At the same time, more Anchor will be generated, which is usually more time consuming This means that part of the reasoning time is sacrificed in exchange for a significant improvement in the detection results As far as the result is concerned, it is worthwhile.

This proves that our improved model can more accurately identify wheat spider mites while maintaining a lower fault tolerance rate. In more detail, we visualize the loss of our model during training, as shown in Figure 7, where the loss has converged when the iter reaches around 60,000.

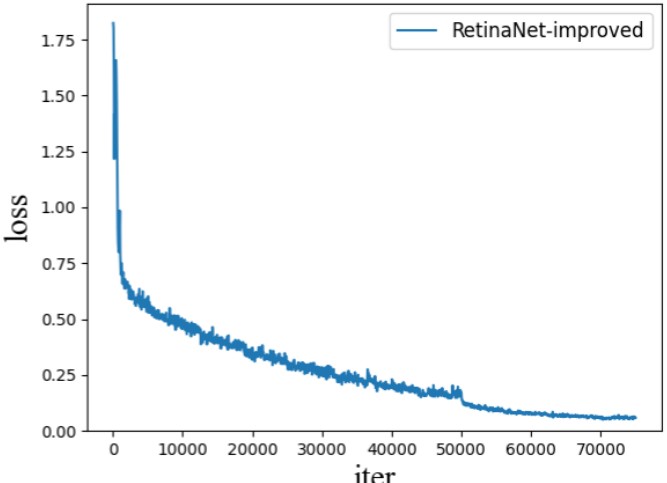

**Figure 7.** Loss curve of RetinaNet.

Meanwhile, as shown in Figure 8, we also visualize the accuracy curves of each epoch of different models, where the *x*-coordinate represents the epoch, while the *y*-coordinate does the mAP. In the figure, the accuracy curve of our improved-RetinaNet is the brown one, which achieves the best accuracy on each epoch compared to other models. This shows that our model can not only successfully match more wheat spider mites, but also have higher accuracy, thus demonstrating the advantages of our model in wheat mite detection.

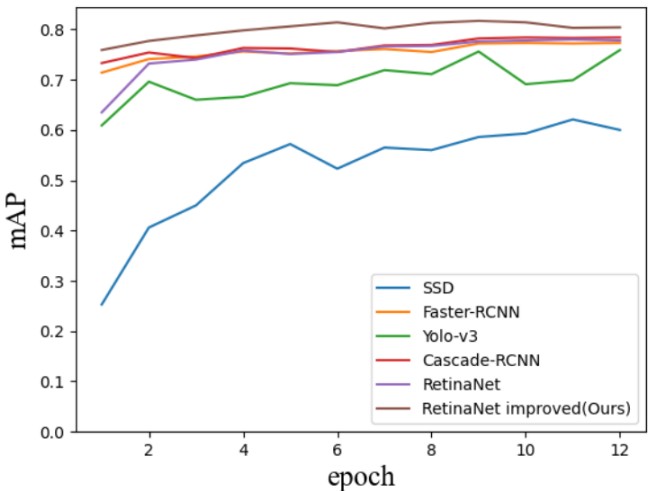

**Figure 8.** Diagram of mAP of different models. The brown is our model.

### 4.4. Different IOU

The Intersection of Union (IOU) is an important index to measure the degree of overlap between the anchor point and the bounding box, and its range is between [0, 1]. The higher the value of IOU, the more overlapping areas and the more accurate the positioning. Different IOU thresholds will affect the effect of Anchor matching. For example, if the IOU threshold is set low, it is difficult to guarantee the quality of the sample. On the contrary, if the IOU threshold is set too high, the number of samples will decrease, resulting in an imbalance between positive samples and negative samples. Therefore, in order to study the influence of different IOU thresholds on the detection results, we also conducted experiments with different IOU thresholds. As shown in Table 3, Recall rate and mAP are still the highest when the IOU threshold is 0.5. Moreover, the more extreme the IOU threshold, the worse the performance of the model. This result indicates that for datasets containing a large number of small objects, a lower IOU threshold does not lead to performance gains.

**Table 3.** The mAP of different IOU.

| IOU | Recall (%) | mAP (%) |
|-----|-----------|---------|
| 0.3 | 87.6 | 77.5 |
| 0.4 | 89.5 | 79.9 |
| **0.5** | **90.2** | **81.7** |
| 0.6 | 90.0 | 80.1 |
| 0.7 | 86.5 | 78.7 |

### 4.5. Ablation Experiments

To further verify the effectiveness of the improved method, ablation experiments are also carried out here. The baseline is the RetinaNet using the ResNet50 backbone. During the experiments, the improved methods are added to the baseline separately. The detection results are shown in Table 4. First, we compare the influence of image segmentation of datasets, as shown in the second line of Table 4. It can be clearly seen that image segmentation is very effective in terms of improving accuracy, with a significant increase of 14.4 %. This shows that image segmentation can not only increase the number of datasets and reduce model over-fitting, but also greatly reduce the serious interference in the image compression process and reduce the loss of image information. As can be seen from the third row of Table 4, a small wheat spider mite detection head can improve the detection performance from 78.0 to 79.8%.

**Table 4.** Ablation experiments on our dataset.

| Model | Image Segmentation | Small Object Head | Context | Anchor-Improved | mAP (%) |
|-------|:---:|:---:|:---:|:---:|:---:|
| | × | × | × | × | 63.6 |
| | √ | × | × | × | 78.0 |
| RetinaNet-improved | √ | √ | × | × | 79.8 |
| | √ | √ | √ | × | 80.2 |
| | √ | √ | √ | √ | 81.7 |

The results show that the model has higher robustness and can better detect the small wheat spider mites. Next, FPN contextual information fusion and multi-scale fusion are added, and the results in the fourth row of Table 4 show that this fusion can improve the feature extraction ability of the model. Finally, as shown in the last row of Table 4, the accuracy of the redesigned Anchor generation scheme is improved by 1.5%, which indicates that the generated Anchor matches the GT better and is more conducive to the model optimization and accuracy improvement.

Besides, we visualize the thermodynamic feature maps of Backbone and FPN, as shown in Figure 9b,c, respectively. Blue indicates very low concerns, while red or yellow

indicates very high concerns. Experimental results show that the model successfully extracts the characteristics of wheat spider mites, which proves the effectiveness and feasibility of the model in detecting wheat spider mites. In addition, the detection effects of our improved model and the original model are compared. The experimental results are shown in Figure 10. For the same image, Figure 10a is the detection result of the Retinaet, and Figure 10b is the detection result of our improved model. It can be seen that the results of our model are better, which can not only can accurately identify the objects ignored by RetinaNet, but also have a higher confidence rate. This illustrates the effectiveness of our improvements. Compared with the original RetinaNet, our improved model enables more wheat spider mites to be recognized successfully which were previously ignored.

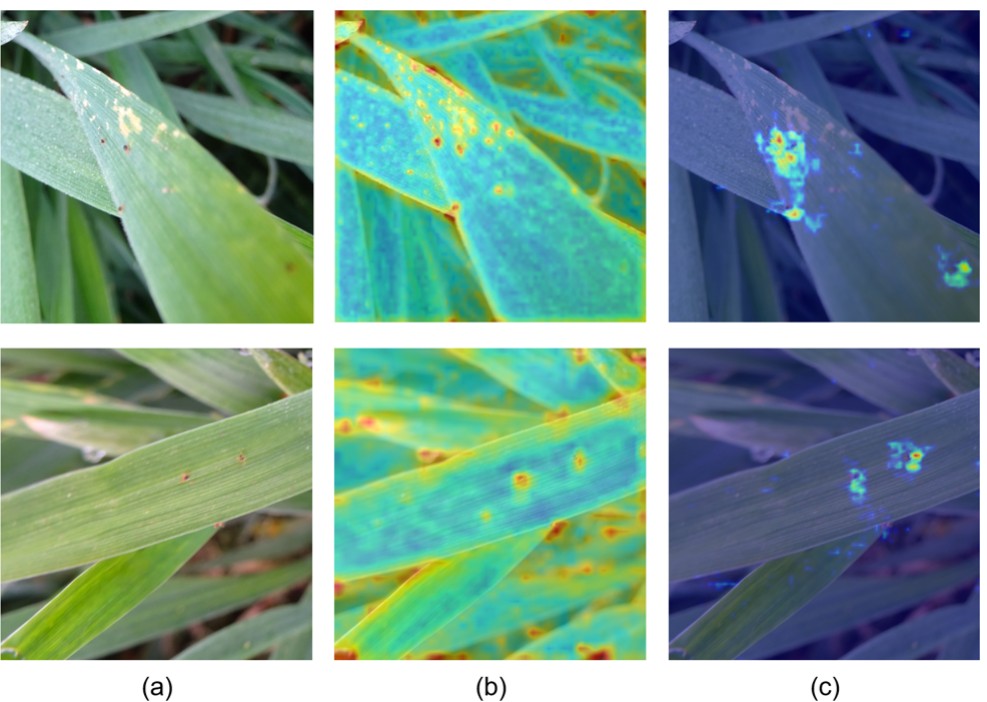

(a)  (b)  (c)

**Figure 9.** The visualization of thermodynamic feature maps of Backbone and FPN: (**a**) the original image; (**b**) thermodynamic map of Backbone (**c**) thermodynamic map of FPN.

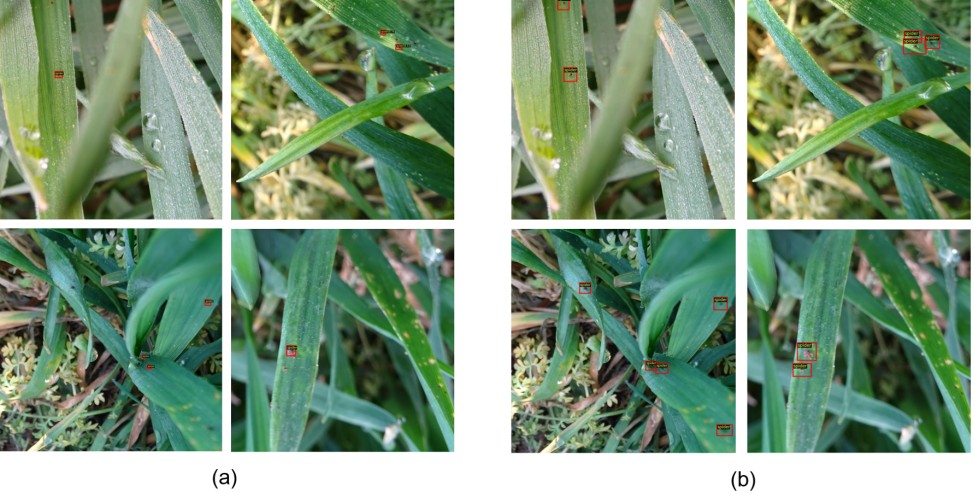

(a)  (b)

**Figure 10.** Comparison of detection effects: (**a**) detection of the RetinaNet; (**b**) detection of our improved model.

## 5. Conclusions

Wheat spider mites are the main harm in the process of wheat growth, which will seriously damage the health of wheat and seriously affect the yield of wheat. However, the small size and complex background of wheat spider mites in real wheat fields greatly increase the difficulty of detection. In order to solve the problem of wheat spider mite identification, a new dataset of wheat spider mites in wheat field background was established in this paper. Furthermore, we use two methods to expand the image dataset. Firstly, the traditional data enhancement method is used to enhance the image dataset. Secondly, the high-resolution images are then cropped into a low-resolution images. Finally, we use these two methods to expand the dataset from 1959 to 9215, which can not only enlarge the data volume, but also alleviate the target minimization problem caused by high resolution to a certain extent. Simultaneously, based on RetinaNet, the model in this paper is perfected to improve its detection efficiency. Specifically, the detection head is added to receive higher-resolution feature maps as input to improve the recognition ability of wheat spider mites. The detection header for small targets can strengthen the context fusion information and significantly reduce the influence of small targets on the model. On this basis, the fusion module of FPN is designed for the fusion of the context information and multi-scale features In addition, our study analyzed that the original anchor generation strategy is not suitable for the detection of wheat spider mites. More importantly, the anchor generation strategy has been improved to make it more compatible with polar wheat spider mites Finally, extensive experimental results show that our method is superior to other advanced methods. However, there are still some limitations. For example, our method increases the amount of computation and parameters. Moreover, the image dataset still needs to be further expanded. In the future work, we will further improve the detection efficiency of the model. At the same time, we will integrate remote sensing data to increase the diversity and multi-source of data. In addition, it is also worth noting that the label allocation is used to improve the matching degree between anchor and Ground Truth.

**Author Contributions:** Conceptualization, D.P. and P.C.; methodology, D.L. and P.C.; validation, D.P. and H.W.; formal analysis, D.P. and H.W.; investigation, H.W.; resources, D.L.; data curation, P.C.; writing—original draft preparation, D.P. and H.W.; writing—review and editing, D.P., H.W. and P.C.; supervision, D.L.; project administration, D.L. All authors have read and agreed to the published version of the manuscript.

**Funding:** This research was funded by the Natural Science Fundation of Anhui Province grant number 2008085QA19, National Natural Science Foundation of China grant numbers 62072002, 62273001, 61906118 and Anhui Provincial Major Science and Technology Special Program grant number 202003a06020016.

**Data Availability Statement:** The datasets generated during and/or analysed during the current study are available from the corresponding author on reasonable request.

**Conflicts of Interest:** The authors declare no conflict of interest.

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
