# Peer review of "Spider Mites Detection in Wheat Field Based on an Improved RetinaNet"

_agriculture, doi:10.3390/agriculture12122160_

Round 1
Reviewer 1 Report
In this paper, the authors constructed a wheat field mite dataset in northern China, and proposed an improved retinal aNet model. First, the wheat mite dataset was expanded from 1959 to 9215 using two different perspectives and image segmentation methods. Secondly, the regression information of the shallow feature map and the semantic information of the deep feature map were fully utilized to further optimize the feature pyramid in FPN. Finally, the generation strategy of anchor was optimized according to the number of mites. However, there are still some problems that need to be noticed.
1. In the Section Experiment, the authors only used two indicators. I am curious about the reason why the authors selected them from many indicators. In addition, wouldn't it be better to verify the advantages of the method proposed by the authors on three or more indicators at the same time?
2. It seems that "The values in bold indicate the best results" is not the same as the authors showed in Table 2. For example, the inference time is the longest, and the number of parameters is not the least. It is suggested that the authors modify the statement here, and at the same time, it’s suggested that the authors analyze the reasons why some parameters produce less superior results, so that the readers can understand the results of this table more clearly.
3. For different IOU thresholds, the authors discussed their influence on the test results. However, the authors didn’t describe the IOU values, but uses it directly. Readers who are not as familiar with this article as the authors will be confused, it is suggested that the authors explain the meaning of IOU and the reason for its value, and then talk about its impact on the network.
4. At the end of the article, the future outlook of the article is too concise. I am curious about the specific improvement direction of the authors in the future and the application direction of the results obtained. It’s suggested the authors to describe more.
5. As we all know, neural networks have a wide range of applications in image recognition and detection. The network architectures used in the following papers also have good performance in similar aims. Besides, in addition to helping expand the dataset, image processing can also facilitate target detection, so it is recommended that the authors cite these papers to expand the description of network or image processing’s importance in related work: “Ensemble meta-learning for few-shot soot density recognition”, “PM2.5 monitoring: Use information abundance measurement and wide and deep learning”, “Deep dual-channel neural network for image-based smoke detection”, “Automatic contrast enhancement technology with saliency preservation”, “No-reference image sharpness assessment in autoregressive parameter space”.
Author Response
Dear Editors and Reviewers,
First of all, let us express our most sincere thanks to the Editor and reviewers. Frankly speaking, without your constructive comments and suggestions, this paper could not have been improved to such a new version. We seriously dealt with the issues raised in your opinions and have made appropriate changes one by one.
According to your important comments, we reply in the attachment.
Please see the attachment!

Reviewer 2 Report
This paper proposed a approaches to detect and locate wheat spider mites by using Retina Net model the proposed measure has the room to be improved before the acceptance of the manuscript.
1.The abstract should reflect the contributions of the manuscript. I suggest rewriting it.
2.Keywords must reflect the core of study same as abstract
3. Introduction should be clearly presented to highlight main ideas and motivation behind the
proposed research. Please include and clearly state research question and motivation of proposed
study in Introduction
4. the authors should analyze how to set the parameters of the proposed methods in the framework. Do they have the “optimal” choice?
5.Section experiment, it would be good to have more information about how experiments have been conducted. What tools/software has been used?
6.It will be valuable to provide some analysis or discussion on the computational complexity for the proposed framework.
7.The novelty of this manuscript should be addressed and emphasized in the discussion section.
8.The conclusion section in the present form is relatively weak and should be strengthened with more details and justifications.
9. Figure captions need to be expanded to make them self-explained.
10.The following papers on the same topic should be cited and discussed:
1. Graph Regularized Nonnegative Matrix Factorization for Community Detection in Attributed Networks
2. Diagnosis of Alternaria disease and Leafminer pest on Tomato Leaves using Image Processing Techniques
3. Robust graph regularization nonnegative matrix factorization for link prediction in attributed networks
Author Response

(The authors gave the same response as above.)

Round 2
Reviewer 1 Report
The authors have carefully revised and improved the quality of the paper according to the comments of reviewers. I think the paper can be accepted and is ready for publishing. I also suggest the authors better adjust the format of the paper to make the whole paper more rigorous. In addition, if the authors can mark the modified content with different colors during the process of revising the paper, it will be better for the reviewers to notice the changes made by the authors more conveniently.
Reviewer 2 Report
I have gone through the revised paper. All my concerns and requests have been carefully addressed by authors.